# AdaptRehab VR: Development of an Immersive Virtual Reality System for Upper Limb Stroke Rehabilitation Designed for Low- and Middle-Income Countries Using a Participatory Co-Creation Approach

**DOI:** 10.3390/bioengineering12060581

**Published:** 2025-05-28

**Authors:** Chala Diriba Kenea, Teklu Gemechu Abessa, Dheeraj Lamba, Bruno Bonnechère

**Affiliations:** 1Department of Information Science, Faculty of Computing and Informatics, Jimma Institute of Technology, Jimma University, Jimma P.O. Box 378, Oromia, Ethiopia; 2REVAL Rehabilitation Research Center, Faculty of Rehabilitation Sciences, Hasselt University, 3590 Diepenbeek, Belgium; teklu.gemechu@ju.edu.et (T.G.A.); bruno.bonnechere@uhasselt.be (B.B.); 3Department of Special Needs & Inclusive Education, Jimma University, Jimma P.O. Box 378, Oromia, Ethiopia; 4Department of Physiotherapy, Faculty of Medical Sciences, Institute of Health, Jimma University, Jimma P.O. Box 378, Oromia, Ethiopia; dheeraj.ramesh@ju.edu.et; 5Technology-Supported and Data-Driven Rehabilitation, Data Sciences Institute, Hasselt University, 3590 Diepenbeek, Belgium; 6Department of PXL—Healthcare, PXL University of Applied Sciences and Arts, 3500 Hasselt, Belgium

**Keywords:** immersive virtual reality, system development, rehabilitation technology, user-centered design, creative thinking, rehabilitation, stroke, low- and middle-income countries

## Abstract

Stroke remains a significant global health challenge, particularly in low- and middle-income Countries (LMICs), where two-thirds of stroke-related deaths occur, and disability-adjusted life years are seven times higher compared to high-income Countries (HICs). The majority of stroke survivors suffer from upper limb impairment, severely limiting their daily activities and significantly diminishing their overall quality of life. Rehabilitation plays a critical role in restoring function and independence, but it faces challenges such as low engagement, limited customization, difficulty tracking progress, and accessibility barriers, particularly in LMICs. Immersive virtual reality (imVR) has shown promise in addressing these challenges, but most commercial imVR systems lack therapeutic design and cultural adaptation. This study aimed to develop culturally adaptable imVR games for upper limb stroke rehabilitation (ULSR) in the context of LMICs, with a particular focus on Ethiopia. The AdaptRehab VR system was developed including six imVR games (Basket Bloom, Strike Zone, TapQuest, FruitFall Frenzy, Precision Pitch, and Bean Picker Pro) through co-creation approaches involving Ethiopian and Belgian physiotherapists, stakeholders, and patients, incorporating game development mechanics in rehabilitation, such as therapeutic aims, cultural factors, feedback, automatic progression recording, task variety, and personalized rehabilitation. It was designed with the Unity 3D engine and Oculus Quest headsets, supporting controllers and hand tracking. This culturally tailored imVR platform has demonstrated significant potential to enhance ULSR accessibility, patient motivation, and outcomes in resource-constrained settings, addressing critical gaps in stroke rehabilitation solutions. In conclusion, the AdaptRehab VR system was successfully developed as a culturally contextualized imVR platform tailored to tackle ULSR challenges in LMICs, with a specific focus on Ethiopia.

## 1. Introduction

Stroke represents a significant global health burden, affecting fifteen million individuals annually and causing over 5.5 million deaths worldwide [1,2]. Notably, two-thirds of stroke-related mortality occurs in low- and middle-income countries (LMICs) [3]. The disparity is further emphasized by the concentration of over 87% of stroke-related disability-adjusted life years (DALYs) in LMICs, a rate approximately seven times higher than in high-income countries (HICs) [4]. This substantial burden underscores the critical importance of accessible rehabilitation services [5].

LMICs face considerable challenges in delivering equitable, high-quality rehabilitation care [6]. The most pressing concerns include a severe shortage of rehabilitation professionals, with some regions reporting as few as 0.5 therapists per 10,000 population, and inadequate infrastructure, particularly in rural areas [7].

In response to these challenges, various international initiatives have emerged. The World Health Organization’s 2030 Rehabilitation Initiatives represent a significant global effort to address rehabilitation needs [8]. Regional priorities are also evolving, as evidenced by health-focused policies such as Africa’s Agenda 2063 [9]. At the national level, countries like Ethiopia have integrated rehabilitation services into their healthcare frameworks, as demonstrated by the inclusion of rehabilitation in both the Health Sector Transformation Plan II and the five-year National Rehabilitation and Assistive Technology Strategic Plan [10]. The recent development and democratization of technology has led to the development of technology-supported rehabilitation. New technology could be a potential solution to increase the provision of rehabilitation services despite the shortage of healthcare professionals, especially in LMICs. Amongst the different technologies, one of the most promising one, for stroke rehabilitation is virtual reality (VR).

VR is a computer-generated system that simulates real-world experiences in a virtual environment [11]. This technology enables the replication of real-world scenarios through specialized VR tools and technologies. The continuous advancement of VR hardware and software has significantly enhanced the realism and immersion of virtual experiences [12,13], creating a sense of physical presence for users within the virtual environment.

Modern VR technologies facilitate the creation of virtual experiences that transcend real-world limitations [14,15,16] and VR’s capabilities extend beyond reality, enabling more immersive and engaging experiences that might be impractical, unsafe, or cost-prohibitive in the physical world [17]. For instance, while traditional rehabilitation carries inherent risks such as patient falls and injuries, these risks can be minimized in VR environments. Furthermore, VR technology effectively addresses challenges in traditional rehabilitation, such as real-time movement tracking and immediate feedback on patient performance and progress [18,19,20].

Immersion, defined as the degree of user engagement within a virtual environment, creates a sensation of presence within computer-generated environments [21]. VR systems are classified into three categories based on immersion levels: non-immersive, semi-immersive, and immersive VR. Among these, immersive VR (imVR) has demonstrated remarkable contributions to healthcare [22,23], education [24,25], and marketing [26,27], with particularly transformative applications in rehabilitation [23,28].

The rapid technological advancements and increased accessibility (due to ease of use and reduced costs) of imVR have led to a substantial increase in its application within stroke rehabilitation. Several recent systematic reviews demonstrate the safety and efficacy of VR as an adjunct to conventional therapy for adults post-stroke, suggesting its potential for routine implementation in upper limb motor recovery [29]. The clinical benefits of various VR modalities for stroke rehabilitation have also been established [30]. Specifically, imVR has been shown to improve function, range of motion, and activities of daily living [31]. Furthermore, targeted improvements in specific joint mobility and function, particularly in the shoulder, wrist, and hand, have been observed depending on the device used [32]. Although imVR can be utilized across all phases of stroke recovery (acute, subacute, and chronic), studies suggest greater efficacy in the subacute and chronic phases compared to the acute phase [33].

imVR has been shown to be effective in rehabilitation by reducing pain, accelerating recovery times, providing realistic environments, decreasing dependence on rehabilitation personnel, enabling increased treatment intensity and frequency, facilitating creative treatment delivery, standardizing therapeutic activities, and improving overall recovery outcomes [34,35]. Key features include real-time feedback [36], customizable and adaptive environments [37], enhanced motivation [38,39], safe and controlled environments [40], and multisensory stimulation including visual, auditory, and haptic feedback [41,42]. Additionally, imVR facilitates essential repetitive exercises [43,44], with numerous studies demonstrating its effectiveness in stroke rehabilitation [32,33,34]. Furthermore, recent studies have demonstrated the feasibility of implementing advanced rehabilitation technologies in LMICs [35,45], suggesting the potential for significant enhancement of rehabilitation services in regions facing professional shortages [45]. However, the implementation of imVR in healthcare, particularly for rehabilitation in LMICs, faces multiple significant challenges. These include limited technical expertise for development and maintenance [35], insufficient multidisciplinary collaboration [45], inadequate government funding [46], and a shortage of rehabilitation services and technologies [47,48,49,50,51,52,53]. Furthermore, existing imVR applications, primarily designed for HICs, often fail to address LMIC-specific needs and local technical realities.

The adaptation of HIC-developed imVR applications in LMICs is hindered by cultural sensitivity issues, proprietary restrictions, and frequent technological updates. These challenges are compounded by inadequate training and awareness of imVR technologies [54], limited research on effectiveness in LMIC contexts [55], and significant cultural and language barriers, as most systems are designed in English [56].

Despite the increasing stroke patient population in LMICs, there are currently no commercially available imVR solutions incorporating local cultural factors [57]. This gap is particularly critical given that 50-80% of stroke survivors experience upper limb impairment [58], significantly impacting their activities of daily living (ADLs) and contributing to perceived disability.

While evidence demonstrates the potential of imVR in rehabilitation, there is a significant gap in research specifically addressing upper limb stroke rehabilitation (ULSR), particularly in LMICs. Current commercial imVR solutions are predominantly designed for recreational purposes rather than therapeutic applications, lacking a scientific foundation and rehabilitation-specific considerations. Additionally, imVR applications developed for specific geographical contexts often fall short in different cultural settings due to linguistic and social variations [59].

To address these challenges, our study aims to develop the AdaptRehab VR system, focusing on designing and developing a culturally adapted imVR platform tailored to support ULSR in LMICs. By employing a co-creative design thinking approach, we ensure that the system aligns with the diverse educational, social, and cultural backgrounds of users, thereby enhancing its effectiveness and relevance in their daily lives.

## 2. Materials and Methods

A human-centered design approach, grounded in design thinking principles (see Figure 1), guided the exploration of stroke survivors’ needs and preferences. Design thinking is a human-centered approach to innovation that emphasizes understanding users’ needs, fostering creative solutions, and delivering practical outcomes [60,61,62]. This collaborative and iterative process included in-depth interviews designed to elicit rich qualitative data regarding the target population’s motivations, challenges, and perspectives related to their limited functional levels and specific rehabilitation needs (see Table 1). This methodology draws inspiration from previous work in design thinking for health applications [63,64] but adapts it specifically to the context of understanding rehabilitation barriers in stroke survivors in LMICs.

A collaborative, co-creation design approach underpinned the development process, uniting rehabilitation professionals, patients, relevant stakeholders, and developers from both Jimma University and Hasselt University (Figure 2). This participatory methodology ensured that the imVR games were culturally sensitive, contextually appropriate, and aligned with therapeutic objectives. Specifically, the development process integrated the expertise of rehabilitation professionals from Jimma University Specialized Hospital and the lived experiences of patients receiving care at the same facility. Patient feedback was gathered through individual interviews conducted after gameplay sessions, providing valuable insights into usability, engagement, and perceived therapeutic benefit. Concurrently, rehabilitation professionals participated in focused group discussions, offering their professional perspectives on the games’ alignment with clinical best practices and their potential for integration into existing rehabilitation programs.

AdaptRehab VR was specifically developed for the Ethiopian context, utilizing the Afaan Oromoo language, widely used by the population in the study area. As one of Ethiopia’s most widely spoken languages, Afaan Oromoo is used by millions across various regions. It holds particular significance as the official language of Oromia, the country’s largest regional state. The choice to develop AdaptRehab VR in Afaan Oromoo was motivated by a commitment to improving accessibility and usability for local patients, healthcare providers, and rehabilitation specialists. By incorporating the native language, the platform significantly enhances engagement, comprehension, and interaction.

This iterative feedback loop, facilitated by the development team, ensured that the imVR games evolved to meet the expressed needs, preferences, and expectations of the patient population. Developers meticulously incorporated feedback from patient interviews, local stakeholders, and professional group discussions, making continuous adjustments to game mechanics, content, and difficulty levels. This dynamic process optimized the balance between therapeutic efficacy and user engagement, creating games that were both functional and enjoyable.

The inclusion of game developers, rehabilitation professionals, and local stakeholders in this co-creation process was crucial for several reasons. First, as highlighted by [65], this collaborative approach is essential for developing imVR games that are not only therapeutically sound but also culturally relevant and readily adaptable to the specific context of implementation. Furthermore, engaging stakeholders from the outset, as emphasized by [66], fosters a sense of ownership and promotes long-term sustainability.

By incorporating their concerns and feedback throughout the development lifecycle, the project ensured that the resulting games were well-aligned with the needs of the target population and poised for successful integration into the local healthcare ecosystem. This early and continuous stakeholder involvement is critical for broader acceptance, enhanced engagement, and effective evaluation of the imVR games’ therapeutic impact.

To assess the acceptance and usability of the AdaptRehab VR system, we employed the technology acceptance model (TAM) [67]. TAM is widely used to understand and predict user acceptance of new technologies by evaluating perceived ease of use, perceived usefulness, and user satisfaction. The TAM survey was administered to both clinicians (*n* = 4) and stroke patients (*n* = 6) who participated in a pilot study. TAM analysis provides valuable insights into the system’s potential for integration into clinical practice and patient rehabilitation routines.

## 3. Results

### 3.1. Game Setting

The selection and design of various assets and objects used in the games were curated to reflect Ethiopia’s cultural and social context accurately. This thoughtful approach ensures that the games resonate with the target audience and provide a meaningful and immersive experience. These games were developed using the Unity 3D engine. The programming was carried out in C#, a versatile language known for its effectiveness in game development.

The developed AdaptRehab VR was specifically designed with the intention of being implemented in hospitals within LMICs. Its development takes into account the unique challenges these regions face, such as limited access to advanced rehabilitation technologies and healthcare resources. The iterative feedback loop and the need for contextual and user-centered design necessitated involvement with the hospital setting.

For creating a VR experience, we utilized the Oculus Quest 2. This device supports both controllers and hand tracking, allowing patients to engage in exercises within a virtual environment, and is widely available and affordable. The inclusion of hand tracking is particularly noteworthy as it enables users to interact with the virtual world more naturally and intuitively, offering a therapeutic and engaging experience that can enhance patient rehabilitation outcomes. It is a standalone VR headset, meaning it operates independently without requiring a wired connection to a PC or external hardware. Unlike traditional VR systems that rely on external sensors or base stations for tracking, the Quest 2 features an advanced inside-out tracking system. This system utilizes built-in cameras and sensors to track head and hand movements in real time, providing a seamless and immersive VR experience without the need for additional equipment. Additionally, its wireless design enhances mobility and convenience, allowing users to enjoy VR anywhere without being restricted by cables (Figure 3).

### 3.2. Codes and Development

The AdaptRehab VR system was developed utilizing a comprehensive suite of cutting-edge technologies. The core components were created using Unity 3D Engine (version 2022.3.43F1) and Visual Studio 2022. Blender and Audacity were used to design 3D models and to record and edit audio clips, which were incorporated into the system for feedback and background music, respectively. Firebase was integrated for data management. The different components are presented in Figure 4.

The codebase harnesses a variety of Unity APIs to implement its game mechanics, therapeutic objectives, and user interactions. Key components include game object, transform (for position, rotation, scale), and various physics components such as rigid body for movement and collisions. The system further utilizes collider types like box, sphere, and mesh for complex shapes, alongside audio components managed by audio source for sound clip and background music integration. User Interface (UI) elements are crafted using components like text, button, image, and slider within the canvas, with Firebase API supporting data operations.

To incorporate Oculus Quest 2’s hand-tracking abilities, the Oculus Quest package was integrated using Unity package manager. This facilitated seamless management of dependencies including XR Origin (XR Rig), XR Interaction Toolkit, Oculus XR plugin, hand tracking (XR Hands), and Open XR plugin. Hand tracking was enabled in collaboration with the hand tracking package, employing computer vision algorithms to detect and render hand movements. Pre-built components like hand tracker and hand renderer streamlined the process, allowing efficient hand detection and rendering within the virtual environment.

The system’s architecture follows a modular design with dedicated scripts, including game controllers and therapy managers, governing game logic and therapy sessions, respectively. Game controller scripts manage game objects and assets, while therapy manager scripts oversee therapy session data and progress tracking.

Asynchronous programming is leveraged through the coroutine class, ensuring a fluid and responsive user experience. This approach allows tasks such as asset loading and user input processing to run in the background, maintaining high performance even in complex scenarios.

Integration of various packages and APIs was achieved through a combination of Unity’s built-in package manager and custom scripting. The Unity package manager handles installation of Oculus Quest and hand tracking packages, while custom scripts bridge functionalities and streamline API integration.

Firebase was seamlessly integrated into AdaptRehab VR games, offering advantages such as user progress tracking, data storage, user authentication, and player behavior analysis. Secure profiles allow patients to log in using patient IDs, facilitating secure data management, multiple profiles, and real-time progress tracking. Healthcare providers can remotely monitor patient progress, access rehabilitation assets, and assess exercise effectiveness. The system captures exercise success rates, session durations, and patient engagement metrics. Additionally, it provides insights into popular or effective rehabilitation exercises and allows plotting of RoM (Range of Motion) values to observe improvements over time. Figure 5 details the Firebase real-time database structure.

### 3.3. Game Design Mechanism

The AdaptRehab VR games portfolio features a diverse array of titles, including Basket Bloom, Strike Zone, TapQuest, FruitFall Frenzy, Precision Pitch, and Bean Picker Pro. Each game is crafted with unique design mechanics that encompass movement dynamics, user feedback, interactive elements, and specific therapeutic goals. These design elements are thoughtfully integrated into the imVR game development process, offering an innovative strategy for rehabilitation.

This approach has the potential to significantly address the challenges faced by LMICs in providing effective rehabilitation services. By tailoring the design mechanics to align with therapeutic objectives, the games not only engage users but also support targeted rehabilitation outcomes.

### 3.4. Game Environments

The AdaptRehab VR system leverages a suite of open-source Unity assets to create engaging and culturally relevant rehabilitation exercises. Each game within the system is designed with specific therapeutic goals in mind, while also prioritizing patient immersion and motivation. The complete detailed explanations of these design mechanics and their therapeutic applications are provided in Table 2 and screenshots of the serious games (SGs) are presented in Figure 6.

#### 3.4.1. Basket Bloom

This game fosters object recognition and sorting skills within a simulated living room environment. The virtual room is constructed using basic geometric primitives (cubes for walls and ceiling) and furnished with common household items such as a table, sofa, and decorative elements such as flowers. This detailed environment enhances immersion and provides a relatable context for the therapeutic tasks. Gameplay revolves around sorting locally familiar fruits and vegetables into color-coded and labeled baskets, reinforcing cognitive skills while incorporating culturally relevant elements specific to Ethiopia.

#### 3.4.2. Strike Zone

This game, designed to improve reaction time and upper extremity coordination, utilizes a simplified aesthetic, employing basic geometric shapes such as cubes game object and balls from Unity asset store. The minimalist design minimizes distractions and allows patients to focus on the core mechanics of the game.

#### 3.4.3. TapQuest

Similar to Strike Zone, TapQuest utilizes a minimalist environment constructed from the cubes game object to facilitate the core game mechanics.

#### 3.4.4. FruitFall Frenzy

Combining elements from Basket Bloom and Strike Zone, FruitFall Frenzy utilizes a variety of fruits, vegetables, baskets, and cubes.

#### 3.4.5. Precision Pitch

This game aims to improve aiming and upper extremity control. The environment consists of horizontal and vertical walls (constructed from cubes) designed to contain balls, along with a table element. 

#### 3.4.6. Bean Picker Pro

This game uniquely incorporates the culturally significant Arabic coffee tree and its beans. Highlighting the cultural relevance of the chosen elements reinforces the tailored approach of the AdaptRehab system.

### 3.5. Customization/Personalization

The system offers adjustable difficulty levels determined by rehabilitation professionals to provide a personalized therapeutic experience. This customization, based on factors such as height, distance, speed, and the number of objects (balls, fruits, and vegetables), allows therapists to tailor the challenges to each patient’s individual needs and progress. Patients can select both the game and the difficulty level based on their preferences and therapeutic goals, fostering engagement and motivation. The difficulty of the exercises in the system is adjusted both automatically and manually, ensuring a user-friendly and personalized experience. The system is designed to automatically adapt the difficulty level based on the patient’s performance. When patients successfully complete a specific level, the system automatically increases the difficulty level to provide a continuous challenge. For therapists who wish to manually adjust the difficulty levels between exercises, the system offers a convenient menu. This menu automatically opens and presents various configuration options for the particular serious game (SG), allowing therapists to customize the difficulty settings according to the patient’s needs and progress. This dual approach ensures that the rehabilitation process is both dynamic and tailored to individual requirements, enhancing the overall effectiveness of the therapy. This adaptability ensures the system can cater to a wide range of abilities and progressively challenge patients as they recover. For example, in “Basket Bloom”, the difficulty could be adjusted by increasing the number of fruits and vegetables to sort, changing the height or distance, or modifying the complexity of the basket labeling. Similarly, in “Strike Zone”, the speed and trajectory of the balls could be altered, along with the number of targets to hit. This flexible framework enables a dynamic and individualized rehabilitation journey.

Table 3 outlines the main clinical characteristics of each game included in the AdaptRehab VR system. This table specifies the stroke stages (acute, subacute, chronic) each game is suited for, the primary rehabilitation aims addressed (strength, range of motion, coordination, fatigue, fine motor skills), and any contra-indications that therapists should consider before recommending a particular game to a patient. This information is crucial for tailoring the rehabilitation program to the individual needs and capabilities of each stroke survivor, ensuring a personalized and effective recovery process.

### 3.6. Technology-Acceptance Model

#### 3.6.1. Perceived Ease of Use

All four clinicians rated the system highly on perceived ease of use, with an average score of 4.5 out of 5. They found the system intuitive and easy to navigate, attributing this to the user-friendly interface and straightforward instructions. The clinicians appreciated the automatic progress tracking and the ability to customize exercises based on individual patient needs. They noted that the system required minimal training, making it accessible even for those with limited experience in VR technology.

The stroke patients also reported a high level of ease of use, with an average score of 4.2 out of 5. Patients highlighted the natural interaction facilitated by hand tracking, which made the exercises feel more like real-world activities. The immersive environment created by the headset was particularly praised for its simplicity and comfort, allowing patients to focus on their rehabilitation tasks without technical distractions.

#### 3.6.2. Perceived Usefulness

The clinicians unanimously agreed that the system was highly useful, giving it an average score of 4.7 out of 5. They emphasized the system’s ability to provide real-time feedback and progress tracking, which they found invaluable for monitoring patient progress and adjusting treatment plans. The clinicians also noted that the system’s engaging and motivating game design could significantly improve patient adherence to rehabilitation programs.

Stroke patients rated the system’s usefulness with an average score of 4.3 out of 5. They found the games enjoyable and motivating, which encouraged them to engage more actively in their rehabilitation exercises. The patients also appreciated the immediate feedback and the ability to see their progress over time, which they felt enhanced their sense of accomplishment and motivation to continue with their rehabilitation.

#### 3.6.3. Overall Acceptance

The overall acceptance of the AdaptRehab VR system by clinicians was very positive, with an average score of 4.6 out of 5. The clinicians saw great potential in the system for improving rehabilitation outcomes and patient engagement. They were enthusiastic about integrating the system into their clinical practice and believed it could address some of the challenges faced in traditional rehabilitation, such as low motivation and limited customization.

The stroke patients also showed strong overall acceptance of the system, with an average score of 4.3 out of 5. They found the system to be a valuable tool for their rehabilitation and were eager to continue using it. The patients felt that the system could help them achieve better functional outcomes and improve their quality of life.

## 4. Discussion

### 4.1. Main Results

In the development of rehabilitation games, aligning with both therapeutic aims [68] and cultural adaptability [69] was crucial to ensure sustainability, effectiveness, and efficiency. The AdaptRehab VR system addresses a range of therapeutic goals, including pain management, improving RoM, strengthening muscles, enhancing coordination and motor control, restoring functional activities, and improving hand–eye coordination. Its primary objectives are to restore affected functions, boost independence, and enhance quality of life. The AdaptRehab VR system was developed through a co-creation process involving rehabilitation professionals, stakeholders, and patients. Feedback from these contributors was integrated at every stage of game development to ensure that the final products met patient needs, addressed stakeholder concerns, and aligned with therapeutic objectives. AdaptRehab VR encompasses six games, each targeting specific rehabilitation goals.

The results of the TAM survey indicate that both clinicians and stroke patients perceive the AdaptRehab VR system as easy to use and useful for rehabilitation. The high mean scores for perceived ease of use and perceived usefulness suggest that the system is user-friendly and meets the needs of both clinicians and patients. The correlation analysis and regression analysis results support the TAM theory, which suggests that perceived ease of use is a significant predictor of perceived usefulness.

Before discussing specifically these six games, it is important to note that goal-oriented dual-task proprioceptive training can be highly effective in LMICs, even without using rehabilitation technologies [70]. Therefore, we incorporate activities aimed at improving upper limb function, such as picking up virtual objects, reaching and grasping, throwing, tapping, collecting, and hitting objects. These activities serve as the primary goals for rehabilitation. Additionally, the games are designed to enhance cognitive function by challenging users to identify objects based on their shape, color, labels, and size, while also promoting eye–hand coordination and the ability to avoid objects that should not be collected or hit. In this way, the AdaptRehab VR system integrates dual-task exercises that target both upper motor function and cognitive function improvement.

Basket Bloom was designed to improve the RoM in shoulder and arm movements. Players engage by grasping and releasing objects into the appropriate baskets, stimulating movement and flexibility. Numerous studies support the effectiveness of these actions in enhancing shoulder and arm functionality, especially in stroke rehabilitation [71,72,73]. Significant improvements in upper limb functionality have been documented in various imVR games designed for object manipulation. For example, patients who engaged in a ball-picking basket game showed statistically significant improvements [74]. Another study demonstrated excellent reliability in moving cubes between compartments in imVR, confirming the method’s effectiveness in upper limb rehabilitation [75]. Furthermore, studies have shown that virtual activities involving grasping and releasing objects can yield better upper limb motor responses compared to traditional rehabilitation [76,77,78].

The hand tracking feature is specially crafted to enhance finger and wrist functionality in subacute patients, integrating all features available in the controller version of the games. Studies have indicated that hand tracking significantly benefits arm and shoulder rehabilitation in imVR environments by facilitating object manipulation. This feature is noted for its suitability across different severity stages of impairment, with studies highlighting its high effectiveness scores. Hand tracking also contributes to making imVR applications more intuitive and immersive for patients, thus improving the rehabilitation experience [79,80,81].

The second game, Strike Zone, aims to enhance the speed and accuracy of arm and hand movements by challenging players to hit incoming balls at varying heights and distances. Multiple studies have shown that imVR games like Strike Zone can improve these motor functions [79,82,83]. This game also incorporates hand–eye coordination exercises that help strengthen the shoulder, with various studies documenting significant improvements in shoulder function through imVR interventions [84,85].

TapQuest focuses on improving finger, wrist, and hand dexterity, emphasizing fine motor skills. Evidence suggests that imVR is effective in enhancing dexterity in these areas [86,87,88]. The game includes activities with cubes appearing at various distances and speeds, encouraging players to react quickly, thereby improving range of motion and hand-eye coordination.

FruitFall Frenzy challenges patients to catch fruits and vegetables using a basket, enhancing gross motor skills. These movements support daily activities such as lifting, pushing, pulling, and throwing, thereby improving shoulder and arm mobility. A study on catching falling stars in imVR games [49] suggested that games like FruitFall Frenzy can motivate patients and be cost-effective for upper limb rehabilitation. Related studies confirmed the potential of such games to enhance shoulder strength [2,3,52].

Precision Pitch is designed to strengthen upper arm muscles through ball-throwing exercises, enhancing the RoM by encouraging reaching and stretching. It includes cognitive elements requiring attention and coordination as patients throw numbered balls at matching targets. Studies [83,89] have shown promising results for similar virtual activities. The hand tracking feature further enhances dexterity by enabling catching and throwing interactions.

The therapeutic focus of Bean Picker Pro is on fine motor skills, particularly pinching and grasping, to enhance finger control. When patients pick beans from a tree, they engage their shoulders, arms, wrists, and hands. The game also incorporates cognitive tasks, requiring patients to identify and collect correctly colored beans, fostering problem-solving skills. Related works in virtual environments have demonstrated their contribution to upper limb rehabilitation [79,90,91].

In addition to the rehabilitation aspects, an important component of this system is the recording of the motion being performed by the patients during the rehabilitation, as these parameters can be used to track the evolution of the patients [92]. It is of note that such technologies could also be used in the future to track and assess other stroke-related disorders such as spasticity, dysarthria, or aphasia when combining data collection with AI analysis [93].

The AdaptRehab VR was developed specifically for stroke patients in LMICs, especially Ethiopia, which faces unique challenges in healthcare access, stroke prevalence, and rehabilitation services [94,95], with the aim of bridging gaps in rehabilitation access. This intervention also addresses the needs of other LMICs, such as Tanzania [96], Burundi [97], Uganda [98], Benin [99], and Kenya [100]. These countries similarly face significant barriers to effective stroke rehabilitation, including limited healthcare infrastructure, a shortage of trained rehabilitation professionals, and insufficient access to specialized rehabilitation services. The increasing stroke burden in these regions, combined with financial and logistical challenges, underscores the urgent need for accessible, cost-effective rehabilitation interventions to improve recovery outcomes for stroke survivors.

By providing cost-effective and accessible solutions in areas where conventional rehabilitation might not be feasible, the system addresses cultural and contextual needs with localized content. This enhances the acceptance and engagement of rehabilitation interventions and ensures they align with the community’s needs. Consideration of linguistic diversity, cultural sensibilities, and specific healthcare practices is key to achieving meaningful engagement and positive healthcare outcomes [101,102,103].

### 4.2. Current Limitations of the System

While the AdaptRehab VR system shows promise as an innovative rehabilitation tool, several current limitations need to be addressed for it to reach its full potential and broader applicability.

The first important limitation is that therapeutic applications of imVR can be significantly impacted by various physical and neurological conditions, such as hemiplegia, spasticity, and retraction [104,105,106]. These factors can hinder patients’ ability to interact effectively with the imVR system, limiting their capacity to engage with virtual objects and move their upper limbs freely. Specifically, the degree of hemiplegia can affect imVR usage in several ways, including restricting voluntary movement, reducing the ability to perform motor tasks, and diminishing engagement and immersion due to physical limitations. Furthermore, spasticity can impair the precise movements required for imVR interaction, complicate the use of standard imVR controllers, and limit rehabilitation exercises by limiting smooth motion. Similarly, retraction can restrict the range of motion, reduce the ability to perform full-body tracking movements and necessitate stretching-focused interventions, possibly requiring assistive hardware.

A second significant limitation is the development of additional games to cater to a wider spectrum of therapeutic goals. Expanding the game library would not only enhance customization options for individual patients but also address diverse rehabilitation needs, accommodating a broader range of motor skills and cognitive functions.

Another challenge lies in the adaptation of the system to accommodate different regional languages within Ethiopia. Given the country’s linguistic diversity, with each region having its own language, localizing the system’s interface and instructions would greatly enhance accessibility and user engagement. This adaptation is crucial for ensuring that patients from various linguistic backgrounds can fully benefit from the rehabilitation tools available.

Additionally, the system requires improvements in features that enable effective follow-up and support for telerehabilitation services. By enhancing these features, AdaptRehab VR could facilitate remote rehabilitation sessions, particularly in rural areas or smaller clinics. This would significantly cut the costs, time, and effort associated with traveling to urban hospitals, thereby making rehabilitation services more accessible in regions where such services might be limited or non-existent due to a shortage of rehabilitation professionals an issue prevalent in many LMICs.

A crucial aspect that remains to be addressed is the conducting of feasibility studies and randomized controlled trials (RCTs). These studies are essential for collecting robust quantitative and qualitative evidence regarding the system’s effectiveness. There is a limited number of studies on VR applications in LMICs, which underscores the need for well-designed research to validate the system’s efficacy and adaptability within these contexts.

Moreover, to foster broader acceptance and utilization, AdaptRehab VR would benefit from enhanced user interfaces designed for ease of use among older adults or those less familiar with technology. Technical considerations, such as ensuring low-cost hardware compatibility and reliable internet connectivity in rural regions, will also play a crucial role in maximizing the system’s reach and effectiveness. Fostering partnerships with local healthcare providers and government-funded initiatives could further facilitate outreach and implementation, ensuring the system meets the specific healthcare infrastructure needs of LMICs. Addressing these limitations can enhance the overall efficacy, accessibility, and scalability of AdaptRehab VR, making it a more comprehensive and versatile tool in the field of rehabilitation.

### 4.3. Challenges of AdaptRehab VR Implementation in LMICs

The development of VR solutions specifically developed for LMICs is only the first step (Figure 7). Implementing VR rehabilitation systems, such as AdaptRehab VR, in LMICs presents a variety of challenges that need careful consideration. These challenges stem from both technological and contextual factors unique to these regions.

One of the primary challenges is the creation of culturally relevant and familiar 3D objects. The effectiveness of VR therapy is significantly enhanced when patients can interact with objects that they recognize and relate to. However, sourcing or creating such 3D objects can be difficult, as it requires either advanced technology for 3D modeling or the availability of skilled 3D design professionals. This necessity drives up costs and requires resource allocation, which might be challenging in LMIC settings.

In addition, there are technical challenges related to developing adaptive game mechanisms. The customization of VR experiences to meet the specific needs and preferences of individual patients is critical for personalized rehabilitation. However, creating such bespoke solutions demands sophisticated software development skills and an intimate understanding of rehabilitation science, making it a resource-intensive process.

Designing a system that is both user-friendly and engaging is another challenge that hinges on graphic design expertise. The visual aspect of VR experiences is crucial in maintaining patient engagement and motivation, especially when the rehabilitation process can be repetitive and monotonous. This requirement necessitates hiring skilled graphic designers who can create visually appealing and intuitive interfaces, which can again contribute to increased costs and resource demands.

Beyond the technical and design-related hurdles, implementation also faces regulatory and economic challenges. Establishing regulatory approval for new medical technologies in LMICs can be a slow and complex process, often due to limited bureaucratic infrastructure or varying standards. Moreover, funding and sustaining the implementation of such advanced systems pose financial challenges, especially where healthcare budgets are already stretched thin. Strategies to overcome these hurdles include forming partnerships with local governments, NGOs, and international health organizations to secure funding and support. Pilot programs and training initiatives could also be instrumental in demonstrating efficacy and building local capacity, thereby creating a sustainable model for VR rehabilitation that could eventually be scaled across regions. Addressing these implementation challenges requires a holistic approach that blends technological innovation with cultural consideration and strategic partnerships.

To support the implementation of the AdaptRehab VR system in LMICs, several strategies can be employed. One approach is to leverage financial support for health innovations from international funding sources such as the WHO, USAID, and the Global Fund. Additionally, scaling up implementation across multiple regions, adapting the technology to local contexts, and investing in the training of healthcare professionals for maintenance and operation are essential for ensuring long-term sustainability and cost-effectiveness. This can be further achieved through collaboration with local stakeholders, health sectors, and government bodies to foster ownership and ensure the system’s continued success.

It is also important to consider that the AdaptRehab system poses challenges for blind patients, as its design could lack the necessary accessibility features to support their participation. Additionally, the audio component may not be suitable for hearing-impaired patients, as they would miss out on essential auditory cues. To ensure the game is fully inclusive, it may be necessary to incorporate alternative modalities.

## 5. Conclusions

This research demonstrates the successful development and potential of AdaptRehab VR, a culturally contextualized immersive VR platform designed to address ULSR challenges in LMICs, specifically Ethiopia. By employing a participatory co-creation approach involving patients, rehabilitation professionals, and stakeholders, we created a system that integrates therapeutic principles, cultural relevance, and user-friendliness. The six engaging, multi-level games target various motor functions, including strength, coordination, and dexterity, while incorporating game design elements like feedback, rewards, and progression to maintain patient motivation. The system’s architecture, incorporating automatic progress tracking, facilitates personalized rehabilitation and enhances engagement for both patients and therapists. AdaptRehab VR offers a promising and innovative solution for improving stroke rehabilitation outcomes in resource-limited settings, potentially bridging the gap in access to quality rehabilitation services. Further research, including feasibility studies and randomized controlled trials, is warranted to rigorously evaluate the system’s effectiveness and explore its wider implementation within the Ethiopian healthcare system and other LMIC contexts. This work represents a significant step towards leveraging technology to address critical healthcare needs in underserved populations and promote equitable access to effective rehabilitation.

## Figures and Tables

**Figure 1 bioengineering-12-00581-f001:**
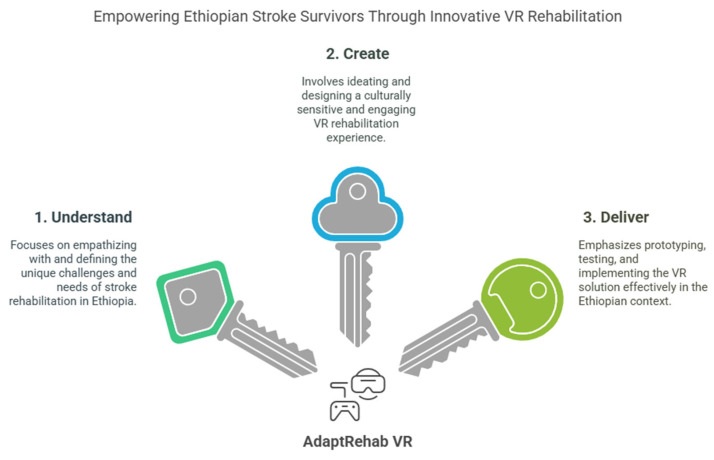
Design thinking principle applied to the development of the system.

**Figure 2 bioengineering-12-00581-f002:**
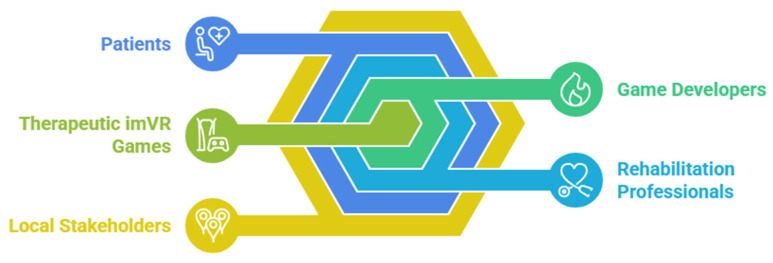
Co-creation design approach used to develop AdaptRehab VR.

**Figure 3 bioengineering-12-00581-f003:**
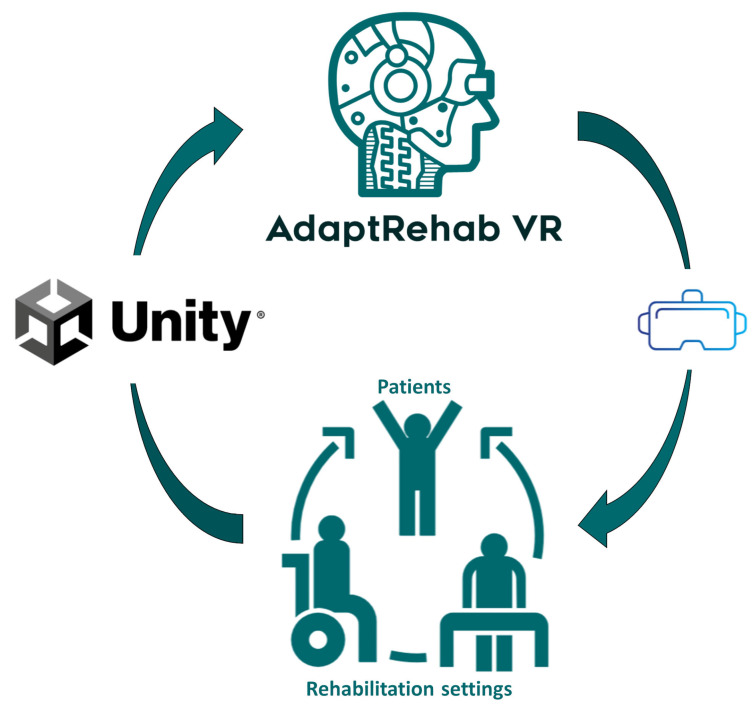
AdaptRehab VR’s stroke rehabilitation games, developed using the Unity 3D engine, utilize a feedback loop. Patient performance data collected by the VR headset informs the system, allowing it to automatically adjust exercise modalities and difficulty levels.

**Figure 4 bioengineering-12-00581-f004:**
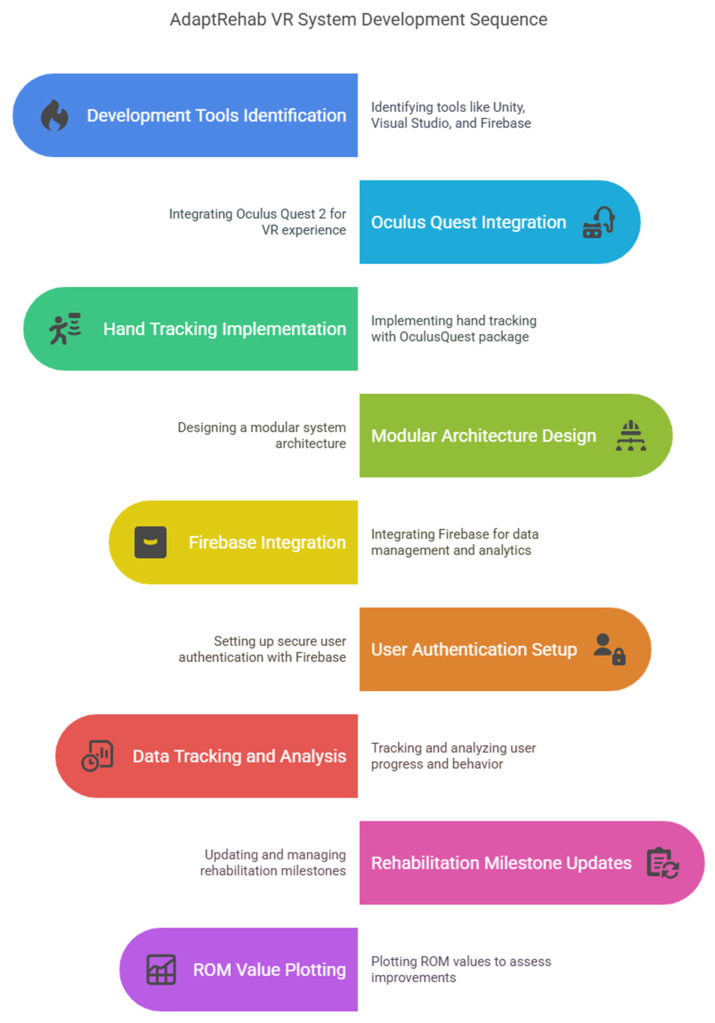
The different components of the systems and flow of system development.

**Figure 5 bioengineering-12-00581-f005:**
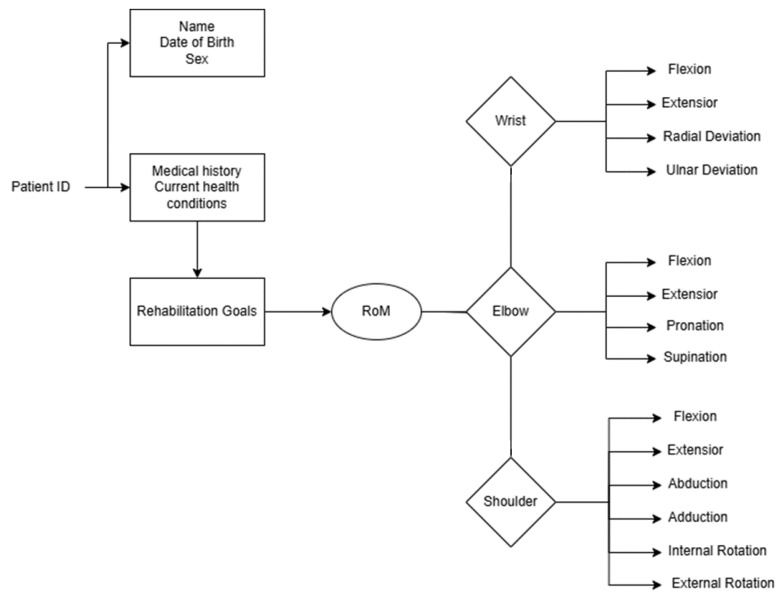
Firebase real-time database structure of continuous data collection, the RoM is used to automatically adapt the game environments and objectives according to real patient abilities and rehabilitation goals.

**Figure 6 bioengineering-12-00581-f006:**
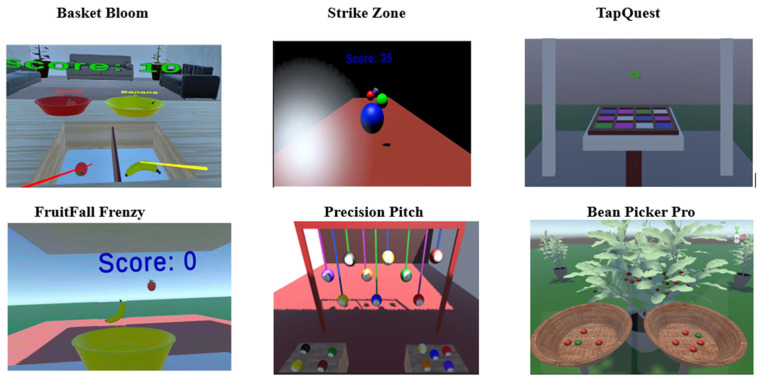
Screenshots of AdaptRehab VR system games.

**Figure 7 bioengineering-12-00581-f007:**
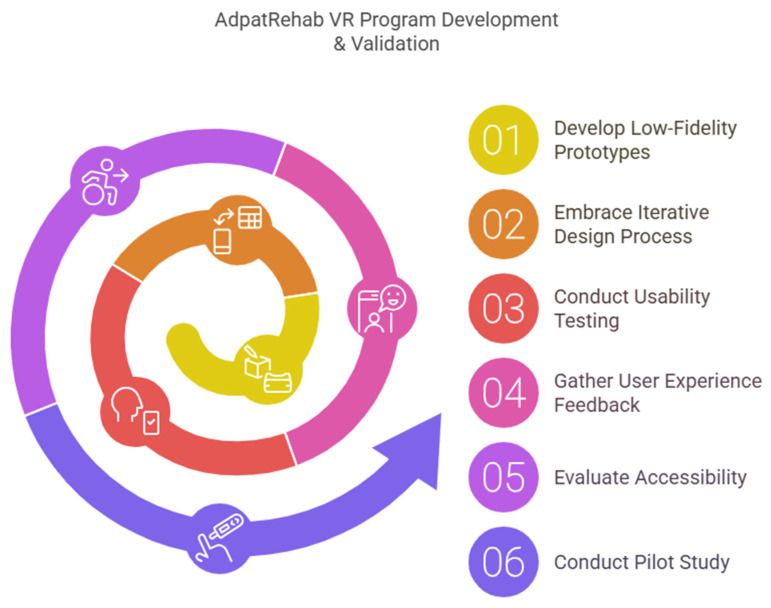
The different steps of the development and clinical validations of the system.

**Table 1 bioengineering-12-00581-t001:** The different steps of the design thinking principles, with main activities and key considerations.

Phase	Step	Activities	Key Considerations
1. Understand	Research	Interviews with stroke survivors, families, therapists, doctors, community health workers Observation of current rehabilitation practices Literature review on stroke in LMICs	Cultural sensitivity Language barriers Socioeconomic context Healthcare infrastructures
	Problem Definition	Synthesize research findings Clearly articulate the core problem(s)	Focus on user needs and aspirations Frame the problem in a human-centered way
2. Create	Ideation and Brainstorming	Generate a wide range of ideas for VR rehabilitation Explore gamification and interactive elements Consider different VR hardware options	Creativity and innovation User engagement and motivation Affordability and accessibility
	Concept Development	Refine and develop promising ideas Design the VR environment and interactions Plan for modularity and scalability	Cultural appropriateness User interface and experience Technical feasibility
3. Deliver	Prototyping	Develop low-fidelity prototypes Create functional prototypes using VR development tools	Rapid iteration and feedback User testing and evaluation
	Testing and Refinement	Conduct usability testing with target users Gather feedback on user experience and effectiveness Iterate on the design based on feedback	Accessibility for diverse users Real-world conditions in Ethiopia Data collection and analysis
	Implementation and Evaluation	Pilot study in Ethiopia Develop partnerships for scale-up Create a sustainability plan	Training and support for users Maintenance and updates Long-term impact assessment

**Table 2 bioengineering-12-00581-t002:** Description of AdaptRehab VR games design mechanics.

Game	Objective	Description	Movement	Feedback	Interaction	Configuration	Therapeutic Aim
Basket Bloom	To reach, grasp, pick, and release fruits and vegetables in their corresponding color-coded and labeled baskets.	Patients grip fruits or vegetables from a box and place them into labeled, color-coded baskets in a semicircle around them. Levels increase in difficulty with more items, baskets, and varying distances.	Moving the upper limb in various directions.	Positive sound and visual effects for correct actions; negative sounds for errors. As well as +5 points for correct placement; −5 points for incorrect placement.	Use controller raycasting to highlight targeted fruits or vegetables and use the grip button to grasp and release them. Hand tracking works in natural ways.	Baskets arranged in a semicircle; color-coded labels for identifications; 10 levels with progressive difficulty.	Improve shoulder, elbow, arm, and hand tracking features to improve finger and wrist functionality. Improve coordination, hand dexterity, balance.
Strike Zone	To hit or avoid balls coming at varying angles, heights, and speeds.	Patients hit incoming balls of various colors delivered from the front, left, and right at different heights and distances. Difficulty increases with speed, quantity, and ball positioning.	Moving arms in various directions, including up, down, diagonal, rotational, backward and forward, circular or arc movements, or a combination of these.	Sound and particle effects when balls were hit. Points are added for successfully hitting balls.	Use controllers or hand-tracking to strike the balls.	Balls delivered at varying speeds, heights, and distances; multiple levels with progressive difficulty.	Enhancing arm and hand movements and hand-eye coordination features exercises to strengthen the shoulder and cognitive challenge. Improve balance and flexibility.
TapQuest	To tap cubes appearing at different locations, distances, and speeds.	Patients tap cubes as they pop up before they disappear.	Moving hands and arms in various directions, including up, down, diagonal, rotational, backward and forward, circular or arc movements, or a combination of these.	Classical local music plays when patients hit the appeared cubes. Points increase based on successful taps.	Use controllers or hand-tracking to tap appearing cubes.	Randomly appearing cubes requiring targeted tapping with hands, fingers, and arms; encourages full arm movements.	Improve finger, wrist, and hand dexterity as well as hand-eye coordination and RoM.
FruitFall Frenzy	To catch and frenzy falling fruits and vegetables using a basket.	Patients catch falling fruits with a basket while avoiding non-catchable objects.	Moving arms in various directions including up, down, and sideways.	Positive/negative sounds and visual feedback for each action. Points were added for correct items, and negative feedback was for missed or incorrect items.	Use controllers or hand-tracking to carry the basket; caught items are visually placed in the basket.	Fruits and objects fall from above; players move and align the basket to catch fruits while avoiding obstacles.	Enhanced shoulder and arm mobility.
Precision Pitch	To throw balls to hit target balls placed at varying distances and heights.	Patients pick up a ball and aim to hit a similar ball at varying heights and positions to the left, center, and right.	Moving shoulders, arms, elbows, wrists, and fingers in various directions.	Sound and particle effects are triggered when targets are hit. Points are awarded for hitting targets.	Use controllers or hand-tracking to catch and throw the balls.	Balls positioned at varying heights and lateral positions; requires precision aiming and throwing.	Strengthen upper arm muscles as well as improve hand, finger dexterity and RoM.
Bean Picker Pro	To reach, grasp, and release beans from a virtual coffee tree and sort them into baskets.	Patients collect coffee beans from a coffee tree using their hands, tweezers, or small scoops.	Moving hands and fingers in different directions.	Visual and auditory responses. Points for correct beans collected while reducing points for incorrect beans collected.	Use controllers or hand-tracking to sort the beans and put them in the basket.	Beans positioned on a tree requiring fine motor skills to grasp; seated configuration promoting upper limb use.	Enhance shoulder, arm, wrist, hand movements, and RoM and strength, particularly the hand tracking feature for fine motor skills and hand dexterity.

**Table 3 bioengineering-12-00581-t003:** Main clinical characteristics of the games.

Game	Stroke Stage			Rehabilitation Aims					Contra-Indications
	Acute	Subacute	Chronic	Strength	RoM	Coordination	Fatigue	Fine Motor	
Basket Bloom		X	X	X	X		X	X	Spasticity Severe weakness
Strike Zone	X	X	X	X		X		X	Severe weakness Poor reaction time
TapQuest		X	X	X		X		X	Severe cognitive impairment Poor reaction time
FruitFall Frenzy		X	X	X	X	X	X	X	Severe weakness Poor reaction time Visual-perceptual deficit
Precision Pitch	X	X	X	X		X		X	Severe weakness Poor reaction time Visual-perceptual deficit
Bean Picker Pro		X	X	X	X		X	X	Severe weakness Poor reaction time Visual-perceptual deficit

Strength: improving muscle power and endurance; RoM: improving range of motion and flexibility; Coordination: improving motor control and coordination; Fatigue: increasing endurance; Fine motor: improving dexterity and precise movements.

## Data Availability

The original contributions presented in the study are included in the article, further inquiries can be directed to the corresponding author.

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
