# Peer review of "AdaptRehab VR: Development of an Immersive Virtual Reality System for Upper Limb Stroke Rehabilitation Designed for Low- and Middle-Income Countries Using a Participatory Co-Creation Approach"

_bioengineering, 2025, doi:10.3390/bioengineering12060581_

Round 1
Reviewer 1 Report
Comments and Suggestions for Authors
This study aimed to develop culturally adaptable imVR games for upper limb stroke rehabilitation in the context of low- and middle-income countries (LMICs), with a particular focus on Ethiopia.
here my comments:
There are no patients involved in the study. It is primarily a description of the use of this software/technology, which could make the article less engaging without a clear clinical framework (see suggestions for the main text).
TITLE: Add the study design to the title.
ABSTRACT:
- Avoid using unverifiable percentages in the abstract due to the lack of references.
- Even if the abstract is unstructured, clearly separate the results from the conclusions.
KEYWORDS:
- Avoid abbreviations.
- Use keywords that are easy to search and relevant to the literature's focus. I don't think that "development" or "co-creation" point to the focus. Consider using MeSH keywords.
INTRODUCTION: Be more consistent in the description of the study's aim.
RESULTS: 1. Explain better how variables, such as different severity of hemiplegia, spasticity or retraction, can influence the use and effectiveness of this technology.
MAIN TEXT:
To increase reader interest and clarify the potential of this technique add its use in other stroke rehabilitation fields, consider adding two important paragraphs:
- The effectiveness of goal-oriented dual-task proprioceptive training (with both traditional and potentially advanced technologies) could be used even without virtual reality (doi: 10.5535/arm.23086) especially in LMICs, but with the addition of this technology the results could be potentially increased.
- Technology could be important also for quantitative and objective assessments and treatment of other stroke-related disorders such as dysarthria or aphasia.
DISCUSSION:
- In the sentence “Developed specifically for stroke patients in low- and middle-income countries (LMICs), especially Ethiopia,” and include a reference that shows that Ethiopia needs more (or other reasons) compared to the other LMICs.
- Also, mention other LMICs considered in the study and their needs.
- When discussing AdaptRehab in LMICs, address its cost, sustainability, and whether funding is available.
- How can LMICs support this cost?
Author Response
See attached document

Reviewer 2 Report
Comments and Suggestions for Authors
The article is devoted to the development of a virtual reality application for upper limb rehabilitation, taking into account the cultural characteristics of low- and middle-income countries. The authors have considered in detail 6 types of exercises created in close collaboration with medical staff and patients. The relevance of creating such applications does not raise any questions, given the problems raised by the authors (lack of localization, cultural context). The description of the developed system, the stages of its creation and the problems of implementation are considered in great detail. The number of links to related research is high, and most of the links are relevant.
I have a few questions about the content:
1. I don't really understand the phrase that the development process took place in a hospital setting. What was meant by that? The testing process? Since the direct development of a VR application can be carried out anywhere.
2. It seemed to me that sections 3.4.2 – 3.4.5 contain the text from the preliminary version of the article. For example, "Further details on the specific gameplay and targeted therapeutic movements would strengthen this description", "A more detailed explanation of the game's objectives and how the environment interacts with the player's actions would improve clarity. " and "Describing the interaction between the player, the balls, and the environment within the game would further enhance this section."- these fragments should be replaced with specific descriptions.
3. Is the difficulty of the exercises adjusted manually or automatically? If there are any calculation formulas for regulating complexity, this could be presented in the text of the article.
4. Has any evaluation of the effectiveness of the developed system been conducted on a small group of participants? Or, for example, did doctors evaluate the effectiveness of the developed system?
5. Is there any feedback from the users of the system? The subjective assessment obtained during preclinical trials or testing? This information could enrich the Discussion section.
Thus, the article is of interest and is highly relevant, but some minor improvements are required.
Author Response
See attached document

Reviewer 3 Report
Comments and Suggestions for Authors
This manuscript addresses a topic of interest for the readership of the journal ad the lack of accessible and culturally appropriate stroke rehabilitation options in low as well as middle-income countries. Moroever, it proposes an innovative solution through the development of a culturally contextualized immersive VR system, AdaptRehab VR. A major strength is related to the human-centered design approach, with the involvement of stakeholders, therapists, and patients throughout the design and development process.
However, the manuscript would benefit from a deeper organization and clarity in some sections. The extensive background, while thorough, borders on redundancy in places and could be more concise. Moreover, while the authors emphasize the participatory approach and cultural relevance, the discussion of evaluation remains limited. The manuscript lacks concrete data or user outcome metrics to support the effectiveness claims, feasibility, usability, or clinical validation.
To maximize the impact of this work, the authors should consider (even a small) a pilot.
Author Response
See attached document

Round 2
Reviewer 1 Report
Comments and Suggestions for Authors
I appreciated the efforts of the authors to improve the manuscript that results clearer and more attractive for readers.
Author Response
Thank you again for taking the time to review this paper and for your useful comments that helps us to improve the quality of this manuscript.
Reviewer 3 Report
Comments and Suggestions for Authors
Thank you for addressing my comments
Author Response

(The authors gave the same response as above.)
